# Brain Synapses: Neurons, Astrocytes, and Extracellular Vesicles in Health and Diseases

**DOI:** 10.3390/ijms27010159

**Published:** 2025-12-23

**Authors:** Jacopo Meldolesi

**Affiliations:** 1IRCCS San Raffaele Hospital, Vita-Salute San Raffaele University, 20129 Milan, Italy; meldolesi.jacopo@hsr.it; 2CNR Institute of Neuroscience, Milano-Bicocca University, 20854 Milan, Italy

**Keywords:** Alzheimer’s, amyloid-β, Arc, astrocyte, axon, biomarket, canonical, cell-to-cell, clathrin, cleft, clinical/therapeutic, dendrite, endocytic, exocytosis, glia, microglia, navigation, neurotransmitter, polyphenolic, pre-synaptic/post-synaptic, reclustered, recycling, special issue, spine, synaptic vesicle, strokes, SubType, tau, trails/impact, triplets

## Abstract

Synapses, abundant in the brain, are structures needed for life. Our Introduction, based on the forms of such structures published few decades ago, helped in developing recent concepts of health and diseases. Growing axons govern their growth by cell-to-cell communication, axon guidance, and synapse orientations. The assembly of synapses requires the organization and function of pre-synaptic and post-synaptic neuronal terminals with a liquid–liquid phase, governed by Ca^2+^ responses of thin astrocyte domains. Upon synapse stimulation, the clefts expand up to several folds while pre- and post-synaptic thickness remains unchanged. In additional responses, neurons co-operate with astrocytes and extracellular vesicles (EVs), the latter dependent on extracellular and intracellular spaces. Astrocyte and microglia cells and/or EV secretions induce neurons by various effects including traveling changes. Pre-synaptic responses are defined as canonical if based on neurotransmitter release; non-canonical if they are without release and are discharged by EVs, not neurotransmitters. Health and diseases depend on other general properties, such as those defined molecularly. Among neurodegenerative diseases, attention is specified by various properties of Alzheimer’s and other diagnoses. Critical identifications can be due to astrocyte and microglia cells or multiple effects induced by EVs. At present, the complexity of therapies, although of limited success, is developing innovative initiatives.

## 1. Introduction

The present Special Issue (SI) deals with synapses existing in the central nervous system (CNS), which have been known for a long time but only recognized by innovated properties identified during the last several years. Considering the developments that take place during the Introduction, I note that at least half of the presented information is due to well-known published events before 2018. A fraction of additional information, published more recently, has been recognized and related directly or connected to previously reported events. Because of the nature of their coordination, the properties presented in this Introduction anticipate recent mechanisms and processes presented in the following sections of this review.

The first step presented here includes the appearance of axons, the first components of synapses including the somato-dendritic compartments of neurons. Such axons are distinct in various properties: composition, diameter, and length [1]. The formation of axons includes progressive extension accompanied by navigation and orientation. Axonal length varies among synapses. Altogether, however, the axons account for high fractions of whole neuronal volumes [2]. In most neurons, the axon is initially single, then organized into multiple branches. Its activity depends mostly on voltage-gated ion channels, necessary for the activity of their action potentials. Membrane trafficking of axons brings membranes and proteins to the plasma membrane, with polarized distribution leading to neuronal subdomains [3].

Widely accepted for many years, synapses are based on the coordination of two neuronal fractions, inducing cell-to-cell signaling events necessary for brain function. Upon their contact, neuron fractions undergo dramatic structural changes, with one becoming pre-synaptic and the other post-synaptic. This connection is the key to synaptic structure and function [4]. The initial processes start from the activation of pre-synaptic terminals, including the exocytoses of their synaptic vesicles (SVs), which are filled by neurotransmitters and distributed to selected sites of the plasma membrane. The discharged neurotransmitters are received by the corresponding receptors at the surfaces of post-synaptic sites. Vesicle exocytosis is followed by membrane retrieval, either the rapid re-uptake or the ensuing recycling of endocytic membranes [5].

The post-synaptic activation of specific receptors occurs at dendrites, the numerous, much shorter, protrusions where signaling activations or inhibitions are received. Dendrites, all covered by spines at their surface membrane, respond by ligand-gated ion channels producing depolarization or hyperpolarization of graded potentials. The essential property of dendrite signaling depends on the possible multiplicity of their plasma membrane rafts covered by specific receptors, appropriately distributed at a close distance from the pre-synaptic discharge. In response to their receptor activation, the post-synaptic signaling grows by the multiplicity, summing up, and circulation of their responses [6,7,8] (Figure 1).

For many decades the synapses, including the two neurons of their cell-to-cell signaling events, have been considered necessary for normal brain function. In addition, neurons have been considered positive for many aspects of diseases in the CNS. At the beginning of 2000, the presence and the role of the extra brain cells, the glial cells, became present and active at synapses. In 2005, the astrocytes, the most important glial cells of the brain, were first recognized as present at the extra role of tripartite synapses [9]. At the same time, they were shown to play roles in synapse formation [10]. In 2010 the astrocyte results were confirmed and extended by the control of synapse activities: establishment, functions, and plasticity [11]. Critical aspects of the astrocyte–neuron interactions, dependent on the structure of astrocytes and the superficial Ca^2+^ responses located in their thin expansions, the leaflets, are presented in Section 4. The mentioned processes have extended the need for astrocytes in most key functions of synapses, including secretion. Moreover, astrocytes are relevant in many cases, not only in health but also in diseases [12,13]. In addition, they are not the only glial cell type involved. In synapses, distinct and highly interesting effects are induced also by the microglia [14].

From the knowledge presented up to now, we can conclude that synapses are produced and governed by neurons cooperating with the two types of glial cells. Although correct, this conclusion is not complete. In the list of actions involved in the generation and functions of synapses we need to include also the extracellular vesicles (EVs), structures generated and released by all types of cells [15]. At synapses, the cells producing EVs are neurons, astrocytes, and microglia. Of the various types of EV in many organs, the type 2 exosome, generated and then accumulated within cytoplasmic multi-vesicular bodies (MVBs), is very abundant in synapses [15]. These exosomes are involved in their generation and in various relevant functions, as such they are highly important in synapses. The synaptic properties of EVs will be discussed in the following sections of this review.

During this Introduction, the properties of synapses, including the two pre- and post-synaptic components, have been presented in sequence, anticipating their coordinate structure and function. In general terms, however, the two main components of synapses have been found to require distinct properties dependent on their innovative developments. Based on these considerations, I have decided to organize my presentation of synapses in six sections. The present review has been focused primarily on the pre-synaptic form, illustrated and discussed in terms of health, diseases, and therapies. In this presentation, various aspects of post-synaptic forms are not yet complete. In the near future they will be reorganized, thus completing the presentation of our whole synapses.

## 2. Axons and Synapses

In the presentation of these synapses, the first investigated areas include neuronal axons, the structures that maintain the pre-synaptic order, necessary for their synaptic transmission [16]. Within all synaptic neurons, axons are peculiar structures characterized by many specific proteins [16,17]. Here I will concentrate on the presentation of the general concepts of axons, with ensuing special interests about synapses. Several general properties of neurons, and their variously observed details, depend not only on axons but also on their multiple, associated EVs. The most important axonal functions include synapse formations, cell-to-cell communications, axon guidance, and synapse orientations [16,17]. Synaptic axons, because of their role in both electrical and molecular signaling, are vital for neuronal operations. Their functions, however, are not always independent. Their interactions with EVs, critical for their physiology, are also very important for many pathological processes [18,19,20].

The studies of synaptic terminals are often carried out by two neuronal terminals: pre-synaptic and various post-synaptic structures, assembled by the third intermediate connector, the thin cleft (Figure 1). Interestingly, analyses of synapses by high-resolution microscopy have revealed the relevance of the three components that carry out functions distinct from each other. In response to cell stimulation, the cleft connectors induce the enlargement of the extracellular space between pre-synaptic and post-synaptic structures (Figure 1) [21]. In such excitatory conditions, the thickness of synaptic clefts undergo increases of several times its original size. These results are exciting because the enlargement of clefts appears as a direct regulator of synapses. In contrast, the thickness of the pre- and post-synaptic structures remains largely unchanged, controlled by space occupied by factors that are not enlarged [22]. The confirmation of exciting results [22,23,24] demonstrates that the changes in clefts, established during stimulation, are due to the activation of protein trees connecting the pre-synaptic to post-synaptic sites. The essential neurotransmitter release and the post-synaptic sites, dependent on rafts of cleft surface oriented in front of the pre-synaptic sites, govern the functional receptors that are at least potentially activated [22,23,24,25].

Another important aspect of synaptic organization and function is the tripartite structure. In addition to the pre- and post-neuronal sites, which are included in all types of synapses, the tripartite forms include a third site, due to astrocyte connections associated with the whole neuronal sites [9,25]. Summing up, therefore, bipartite synapses depend on the two pre- and post-neuronal sites. For the tripartite synapses, the bipartite include also the third portion composed of astrocytes.

In addition to synapses based on bipartite and tripartite sites, the latter including astrocytes (9), others exist which in contrast include also cytoskeleton-associated proteins, the activity-regulated, cytoskeletal–associated protein Arc [26]. In stimulated conditions, the Arc concentration is higher and more active in the post-synaptic than in the pre-synaptic sites, contributing to both and to the three integrated sites [26]. The same Arc protein participates in the distribution of a misfolded form of the tau proteins, active both in the cytoplasm and at the surface of neurons. In the established Arc-dependent release, the EVs play significant effects of neurodegeneration by the elimination of intracellular tau together with the increase in extracellular tau, the latter being accumulated at neuronal surfaces [27].

## 3. Astrocytes, Other Glial Cells, and EVs

The results reported in [18,21] are focused on synaptic fusions due to exocytoses of a neuronal nature, including the pre-synaptic neurotransmitter release and the specific post-synaptic-activated receptors. Because of its two trans-synaptic components, these processes are traditionally called bipartite. Two additional aspects of these synapses have been emphasized during the last decades. The first deals with astrocytes. These cells are always present, essential for the astrocyte functions including the tripartite synapses (9). In addition, a few astrocytes are protective, able to restore synaptic functions which have been previously altered by establishment of their damages. Some of these effects have been found to depend on molecules secreted or released by astrocytes [28,29] (Table 1 in Section 5). In other conditions, the interactions of glial cells with synapses have been shown to take place upon secretion of proteins released not only by astrocytes but also by microglia activation [30]. Other mechanisms are similarly operative in synapses dependent on cytokines, a form of regulator controlling general responses [31]. In the brain, the effects of microglia take place not only at the synapses, but also at other processes, including neuronal development [32] and neuronal circuits [33].

So far, I have emphasized the most important effects induced by astrocytes, operative with receptor agents, regulating the activity of pre-synaptic terminals. In other conditions, however, such effects are induced not only by various types of cells already mentioned, but also by some of their EVs addressed primarily to synapses. The functional EV signaling, released by neurons and glial cells, requires relevant targets of cells achieved in the nervous system [34,35,36]. In other conditions, the EVs, released by their cells of origin, can be implicated in short- and long-distance traveling, operative from either local brain sites, to distinct large areas of the body [34,35,36]. Summing up, in these conditions the distinct morphology of neurons is maintained for extended periods of adult life. The mechanisms that maintain the morphology of mature neurons depend, at least in part, on elevated release of glial EVs [36,37].

The brain areas affected by data from both neurons and glia, spread in the whole CNS, are quite variable. In some cases, the synapses induced by astrocytes are relevant for image recognition [38]. In other cases, they are followed by ischemia [39]. The EVs generated by both neurons and astrocytes are spread in the CNS. The integrated data in terms of signaling are still interpreted as canonical and non-canonical processes of neurotransmitter release (Figure 1). These properties have been established in the course of discussion events occurring at synapses [40,41,42,43].

Almost all the brain’s important processes include neurons and glial cells working together with various types of EVs. Among these processes are the fusions of EVs either with specific neurons, such as those at synapses, or with neuronal diseases of various types [35,38,41,42,43]. Neurodegenerative diseases, occurring after EV treatments, are presented in Section 5.

## 4. Mechanisms of Healthy Synapses

This section is focused on complex functions of important synaptic components and processes active in the brain. The first deals with the specific pre-synaptic responses taking place by unique mechanisms, now defined as liquid–liquid phase separation (LLPS), accumulated in response to multiple electrical forces [44]. Under these conditions, specific proteins involved in neurotransmitter discharge from SVs are specifically confined to small membrane areas of nerve terminals, in which LLPSs are induced by the accumulation of aliphatic alcohols. The ensuing local accumulation governs the accurate distribution by clustering and recycling active zones that support the evoked, Ca^2+^-dependent fusions of SVs followed by the immediate discharge of their neurotransmitters. These results indicate that LLPS, once accumulated in the pre-synaptic terminals, induces relevant processes of such distribution. The ensuing processes include the expansion of clefts, the extracellular spaces between pre- and post-synaptic terminals governed by enlarged trees [21,24] (see Figure 1), and the rapid SV discharge. In contrast, the process spares any expansion of spontaneous neurotransmission [45,46].

Neurotransmission is sustained by the discharge of SV neurotransmitters taking place at critical surface sites of pre-synapses. The SV discharge takes place by fusions, followed by rapid pore closures occurring at the plasma membrane. By these, and additional synaptic processes, the discharged SVs undergo very fast refilling and then re-clustering [47,48,49]. Additional developments have been established about the relevance and function of soma and integration of synapses at significant distances from their sites of origin. Finally, the SV locations and their functions by neurons and activated glia are continuously integrated by ongoing EVs released to the extracellular space [49]. Years ago, the pre-synaptic functions of SVs had been defined by two properties: canonical, due to the co-localization of their SV exocytoses by neurotransmitter release, and non-canonical, due to the lack of neurotransmitter release [50]. Additional non-canonical responses have been recently interpreted based on EVs, first developed during pre-synaptic functions, then released to the extracellular space (Figure 1) [43,50].

In previous sections the pre- and post-synaptic roles have been attributed to neurons, with glial cells mentioned only in relation to their release of EVs. The coordinate interaction with astroglial cells are, however, essential for the synaptic function. Extensive studies of recent decades have revealed details about the coordinate dynamics of nerve terminals interacting with detailed astrocytes [51]. The nuclear center of the latter cells directly continuous to the intense complexes of their branches (also called shafts), emanating to extensive thin leaflets; their connections are somewhat homologous to the spines of dendritic branches. In astrocytes, critical distinction of branches and leaflets is based on their different expression of cytoplasmic organelles, including those of Ca^2+^ regulation, that in the first are abundant, but in the second are lacking [52,53,54], except for minuscule endoplasmic reticulum saccules of Ca^2+^, which are located closely proximal to synaptic terminals [55]. Because of their unique distribution, the leaflets play critical roles of synaptic activation [55,56]. In addition, the leaflet motility influences the ionic signaling and homeostasis of active synapses [57]. Because of these properties the intense integration of neurons and astrocytes at synapses occurs at various spatiotemporal scales [55]. Each leaflet widely expresses connective gap junctions, with established continuities with other leaflets and cytosolic branches. Leaflets governing the local Ca^2+^ dynamics integrate the inputs from neurons by enwrapping over 90% of synapses [55]. This section has illustrated various components of the pre- and post-synaptic functions of healthy cells. The next section extends to pathological stages, including those of diseases and therapies.

## 5. Diseases and Therapies

The presentation of many synaptic processes has been focused on in their healthy state. However, their dependence on pathological conditions has often been mentioned. Here I will recognize such status in various examples of diseases, also introducing some of the mechanisms by which their pathology has been generated.

### 5.1. Diseases

CNS pathology is often due to the high frequency of synapses, of great and critical importance. Here I will start with a few examples concerning important diseases. Several treatments are shown to be administered to various patients depending on the properties of their diseases: psychiatric diseases [58]; many diseases appearing in response to CNS [59]; alterations of neuronal morphology [36]; and biomarkers developed from CNS diseases [60]. In terms of treatments, these diseases affect considerable numbers of patients, however not all of them occur in humans. Our first presentations of the pre- and post-synapses/astrocytes have been shown to induce various forms of disease. For example, astrocytic results of Ca^2+^ signaling [61]; retractions of astrocyte leaflets inducing enhanced fear memory [62]; and altered astrocyte–synapse interactions contributing defects of AD disease in a mouse model [63]. Additional forms of disease processes are presented in [64], a review from a Handbook of Clinical Neurology introducing the synaptic neurons interacting with astrocytes and microglia. EVs involved in these diseases are generated by both types of glial cell. The properties of astrocytes and microglia, with their EVs [65,66] extend to diseases concerning the health information anticipated in the Introduction [12,13,14,15] and in several other cases.

Glial cells, all highly important in the present section, but especially the astrocytes [67] (Table 1), are excellent coordinators employing important chemical properties. Highly important for these studies has been the developments of glypicans, a family of chemicals extensively employed in cancer therapies [68] also recognized to operate with synapses. From 2012, glypican 4, secreted by astrocytes and associated with neurons, was found to act on synapses of the hippocampus; its similar factor, glypican 6, was found to act on those of the cerebellum [69]. Employed glypicans have been found to be more effective when associated with heparan sulfate, inducing effects on synaptic developments, neural plasticity, and neurological disorders, with potential effects on synapses. For example, neurocan, a protein secreted by astrocytes, has been found to induce inhibitory synapses and strong regulation of their formation [70]; a new form of glypican 5 regulates synapse maturation and stabilization [71]; and astrocyte regulation governs synapse formation, maturation, and elimination [13] (Table 1).

**Table 1 ijms-27-00159-t001:** Effects of astrocytes at synapses. Astrocytes are present at most sites of synapses appearing in this review. In three of our sections, various types of astrocyte presentation are highly important. In particular, the following:

Section 1. Introduction.
In 2005, the effect of astrocytes was shown many times: accumulation at the post-synaptic site of tripartite synapses [9]; signaling between glia and neurons; focus on synaptic processes to investigate their plasticity [10]. In 2010 the effect of astrocytes was influential on synaptic formation and function, plasticity, and elimination [11]. Recent evidence has confirmed the astrocyte effects on their secretions [12,13].
**Section 3. Astrocytes and other Glial Cells.**
Important effects of astrocytes, revealed in recent studies, include protection and restoration of synapses induced by vesicle release and/or by astrocyte protein secretion [28,29].
**Section 4 and Section 5. From Mechanisms to Diseases and Therapies.**
Synaptic changes dependent on astrocytes and their EVs [28,29,34,35,36,37,38,39,40,41,42] induce Ca^2+^ effects analogous to those already reported in other pathways, reported here and in the previous sections. Highly interesting are the astrocyte results of recent studies based on the unexpected effects induced by astrocyte-complex structures such as the following: 1. liquid–liquid phase separations in response to multiple electric forces [44,45,46]; 2. unique complexity and regulation of local physical dynamics; and 3. mini-Ca^2+^ responses located at small sites of astrocyte thin leaflets near ongoing synaptic responses [55,56,57]. Effects induced by distinct, astrocyte-secreted forms of factors appearing in various areas of the brain may contribute to the development of diseases [70,71,72,73,74,75,76]. Advancing chemical forms are considered for drug targeting and effects of therapy [64,76,77,78,79,80].

Neurodegenerative diseases, including Alzheimer’s (AD), amyotrophic lateral sclerosis, Parkinson’s, and Huntington’s, have been defined. These diseases, different from each other, include some common properties. Here, interest is reported about flavonoids, a family of polyphenolic chemicals. Because of their neuroprotective function, flavonoids have attracted a lot of interest [72]. Their effects induce reduction in neurodegenerative symptoms accompanied by prevention of synaptic loss and enhanced cognitive functions [70]. A lot of research about these diseases are expected in the future [71]. Only a small fraction of patients, affected by genetic defects, exhibits forms of neurodegenerative diseases at a relatively young age [73]. In most other patients, AD appears after 60 years of age. Among neurodegenerative diseases, advanced results have been recently reported: the identification of genes with the potential role of synaptic vesicles [74]; amyloid-β labeling of synapses in live human brain slice cultures [75]; and the correlation of glial reactivity with synapse dysfunctions across aging [76]. Advancing amyloid-β implications [77] and computational strategies and strong innovations in the treatment of AD diseases [78] are important to advancing AD therapy.

### 5.2. Therapy

Interest in therapy is considerable in the whole of Section 4 and Section 5, directly emphasized in several previous publications, such as [47,48,49,50] and others. The present subsection is dedicated to specific developments of AD and other neurodegenerative diseases. Many developments of these diseases have been reported recently. The advanced forms, first modified by reinforcing their addresses, are loaded by diseases and then released to their specific targets. Good answers and limitations by brain EVs have been reported to induce appropriate results [40,64]. Intense studies in the CNS have been dedicated to in 5 as reported here clinical and therapeutic potential. The latter includes ongoing developments and advanced methodologies employed in AD studies [73,74,75].

Here I intend to mention examples of recent developments in the field of therapy. The first presented studies resulted in aging and therapeutics investigations of patients [76] followed by the development of EV drug delivery systems used to enhance neurological therapy [77]. Three following articles deal with other EVs involved in emerging development of neurodegenerative diseases, presented here in clinical terms [78,79,80], followed by a clinical implication about neurodegenerative diseases [81]. In addition to the AD data presented here, information about therapy is given by reference to other neurodegenerative diseases. The last three publications of this subsection [82,83,84] deal with clinical implications about therapeutic agents different from AD: Huntington’s [82], ALS [83], and multiple sclerosis [84].

## 6. Conclusions

The present study, concentrated preferentially on the pre-synaptic formation and function, depends on general aspects and various details discovered or simply identified during recent studies, concerning neurons, glial cells and EVs [10,11,12,13,33,34,35,36,37,38] (Figure 1). Thick and thin components of external astrocyte complex are presented in Section 1, Section 3 and Section 4 and mentioned on Table 1 of this review.

Main components of axons, characterized by general specificities, play key roles in the development of neurons, the initial components of synapses. Properties of these axons depend on two mechanisms: 1. Pre-synaptic structure; 2. Synaptic cleft thickness [37,38,39,40,41,42].

The effects of EVs, released by neurons and glial cells, are due to various functions: 1. Cell-to-cell and neuron-to-glia communications, axon guidance, and synapse orientations [16,17,44,47,56]; 2. Tripartite synapses, induced by astrocyte addition to bipartite synapses [9,52,54]; 3. Ischemic strokes, traumatic brain injury, intracerebral hemorrhage, and other CNS diseases [47,50,59,60].

Highly important synapses are needed for active mechanism operations on distinct areas of diseases. Attention is dedicated to various forms of neurodegeneration treated by polyphenolic agents of innovative treatments [65,66,67,68,69]. The final innovative therapies are considered of potential interest for relevant, innovative improvements [76,77,78,79,80,81,82,83,84].

## Figures and Tables

**Figure 1 ijms-27-00159-f001:**
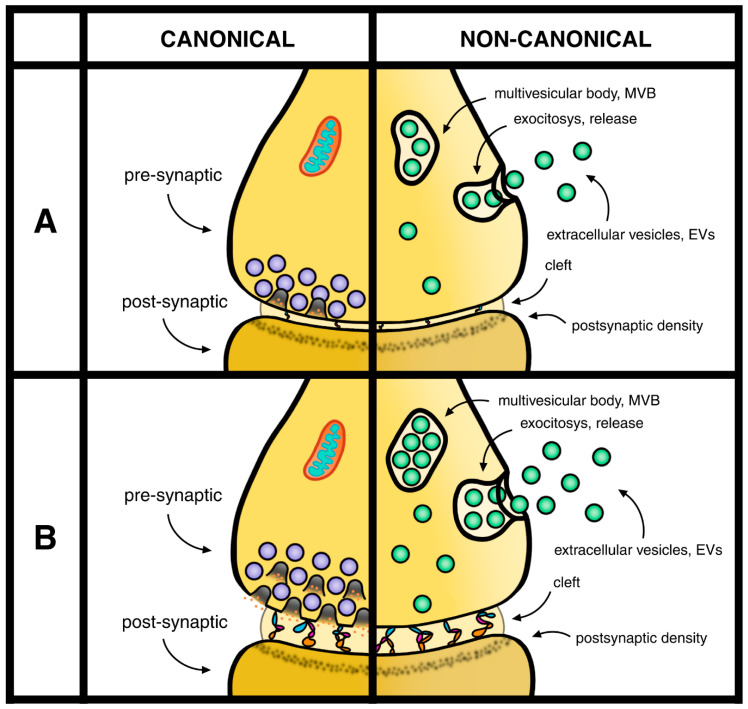
Single pre-synaptic terminal combined to the top of the post-synaptic terminal by a common cleft. The canonical versus non-canonical properties mentioned here refer to separate images of a single neuronal pre-synaptic terminal where two types of vesicles are released by distinct types of exocytosis. This figure includes changes introduced in Figure 1. (**A**) On the left: Together with other organelles (mitochondria and others not illustrated), a group of secretory vesicles (SVs) appears loaded by their neurotransmitter (sky-blue), with a few showing ongoing discharge by exocytosis, while the others wait for their turn. On the top right: Larger exocytic vesicles (EVs, green) are trapped within an endocytic cisterna, the multi-vesicular body (MVB). Upon the exocytosis of MVB, all their loaded EVs, equipped with their membrane, are discharged to the extracellular space. Additional EVs, spread in the cytoplasm, may also be released by exocytosis. Both (**A**,**B**) refer to different conditions of the pre-synaptic terminal connected to the post-synaptic terminal by their cleft. (**A**) During weak stimulation, clefts are thin. (**B**) During strong stimulation the pre-synaptic and MVB exocytoses are stimulated and the cleft space enlarges, connected by thicker trees. During such stimulation the pre-synaptic discharge of SVs is increased, yet the thickness of the pre-synaptic terminal does not enlarge significantly.

## Data Availability

No new data were created or analyzed in this study. Data sharing is not applicable to this article.

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
