# Peer review of "Brain Synapses: Neurons, Astrocytes, and Extracellular Vesicles in Health and Diseases"

_ijms, 2025, doi:10.3390/ijms27010159_

Round 1

Reviewer 1 Report

Comments and Suggestions for Authors

It is with great regret that I suggest rejecting this review. I would like to emphasise that I deeply respect the author of this work, Professor Jacopo Meldolesi, and acknowledge his remarkable contributions to neuroscience worldwide. However, I find that the writing style of the current review “Brain synapses: neurons, astrocytes and extracellular vesicles in health and diseases” does match the style of other MDPI reviews.  

Here, I provide several examples from the text. Abstract: “Synapses, abundant in the brain, are structures needed for life. Our introduction, based on forms of such structures published few decades ago, helped introducing recent concepts of health and diseases. <…>Healthy and diseases depend on other general properties, such as those defined molecularly. <…>Critical identifications can be due to astrocyte and microglia cells or multiple effects induced by EVs. At present the complexity of the therapies, although of limited success, is developing innovative initiatives.”

Introduction: “The present Special Issue (SI) deals with synapses, in particular about these structures, especially those existing in the central nervous system (CNS), with innovations developed during the last several years. Taking into account the developments that take place during such Introduction, I noted that at least half of the presented information is due to well-known events published in early years, i.e. before 2018. Large fractions of the additional information, published more recently, has been recognized directly related or connected to previous events. Because of the nature and of their coordinations, the properties presented in this Introduction appear to anticipate the mechanisms and processes that are presented in the following Sections of this review.”

I suggest either re-writing the text, or changing it from Review to Letter to the Editor/Editorial.

I suggest recruiting other (additional) reviewer to have 2nd or 3rd opinion on the text of review.

Comments on the Quality of English Language

The English could be improved to more clearly express the research. 

Author Response

The evaluation by this Reviewer was limited, focused only on the title and the introduction of my review. This was not a professional evaluation of my review, therefore no report of this evaluation has been presented.

Reviewer 2 Report

Comments and Suggestions for Authors

This paper demonstrates the author's profound knowledge of the subject. This is undoubtedly a review worthy of discussion, but it is highly questionable whether the authors intended it for beginners or experts. There are numerous omitted terms and ambiguous expressions; therefore, we would like them to consider revisions to make it a better review.

Regarding my comments below, I may lack experience; therefore, if the author chooses not to revise them, it is acceptable.

The abstract of this paper contains many abbreviated sentences, likely because of character limits. It feels vague, and some parts are difficult to understand, leading me to think that a more explanatory style would be preferable for beginners. After reading the main text, I think readers will understand what is being said when they read the summary, but I also feel like it does not quite fulfill its function as a summary. Additionally, I feel that the writing style makes it difficult to grasp what the author is trying to convey in this paper. The presentation of information appears somewhat unfocused and scattered, making it difficult to discern what the author is prioritizing. Rather, I felt it would be better to incorporate as many of the details that reinforce the author's intent into the main text itself and to focus the abstract primarily on the author's overview of the field and future prospects.

It might be better to define the abbreviation “EV” when it first appears in the text.

The structure of the sentence in lines 16-17 makes it difficult to grasp the author's intent.

What is the purpose of including this explanation in lines 19-21? While one might anticipate this as an example of synaptic changes, it would be better to make this clearer.

It is unclear what the “other” in lines 21-22 is being compared to.

Overall, the length of the space after sentences seems inconsistent. (e.g., lines 74 and 80, etc.)

Please verify whether the phrase ‘after EV treatments’ in lines 193-4 accurately reflects the authors’ intent. It seemed somewhat abrupt and did not seem to connect with the next section.

The text starting from 263-4 is rather abrupt; therefore, I recommend including an introduction that clarifies the significance of flavonoids in neurodegenerative diseases. The author may have reasons for emphasizing polyphenols, but the explanation does not seem to align well with the direction of this review. As far as I know, many of these substances inhibit protein aggregation, such as Aβ, so the connection to the points discussed in this paper seems indirect and difficult to understand. It would be better to provide more specific explanations related to the review points, such as the treatment mechanisms for neurodegenerative diseases.

This paper contains numerous instances where the descriptions could be more concrete. Given their abundance, we cite only one example.

Sentences like “Good answers and limitations by brain EVs have been reported to appropriate results” exist, but it remains unclear what exactly constitutes “Good answers,” “Limitations,” or “appropriate results.”

When such ambiguous terminology appears excessively, it lends an overall lack of clarity to the writing. In my opinion, descriptions should be written as concretely as possible. However, if the author's intent is to provide an overview of the field as a whole, with details left to other papers, then leaving it as is might be acceptable. Personally, I felt that it could benefit from more careful editing, but I will leave that to the author.

Comments on the Quality of English Language

There are a few instances where the syntax is difficult to understand, so using simpler, more straightforward expressions would likely make the text accessible to a broader readership. Please note, however, that this is the opinion of a reviewer whose native language is not English.

Author Response

In his/her initial evaluation of my review, the present Reviewer has emphasized the importance of synapses, demonstrating the relevance of its general strategy. The structure of my work is not superficial. It does include the literature about synapses focusing on only highly important aspects. I appreciate that this Reviewer, anticipating his/her limited experience in the field, recognizes that my contribution is of general importance. In case the evaluation of the other Reviews is positive, he/she will also agree. Various criticisms have been raised about various parts of my review. Their summary is the following.  

  1. Several sentences about synapses, distributed in various areas of the review, are difficult to understand. I have considered all these sentences and re-expressed them in shorter/simpler forms.
  2. Some abbreviations introduced in the text need to be explained “when they first appear in the text”. I have introduced the requested changes. It should be also emphasize that all the abbreviations appearing in my review are presented and explained on page 1.
  3. The presentation of polyphenols is not clear. I have introduced in the References list an important article concerning the activity of these drugs on cancers that have been reported in the literature before the studies on synapses.
  4. The terminologies employed to account for the mechanisms of action of drugs or factors on synapses, found incomplete or confused, have been characterized.
  5. The English presentation has been systematically corrected also with the collaboration of an American PhD student active in our University.

Reviewer 3 Report

Comments and Suggestions for Authors

This review provides a broad overview of neuronal synapses and their interaction with astrocytes and extracellular vesicles (EVs), with emphasis on their relevance to neurodegenerative diseases. The topic is of considerable importance, and the author’s long-standing expertise in synaptic biology lends credibility. However, while the manuscript is comprehensive in citation scope, it lacks analytical depth, mechanistic integration, and conceptual coherence. The review reads as a chronological compilation of published findings rather than a critical synthesis that advances understanding.

Some major concerns:

1, The manuscript largely reiterates prior findings without critically integrating how neuronal, astrocytic, and EV-mediated mechanisms intersect functionally or pathologically. A stronger mechanistic synthesis, linking EV cargo composition to synaptic modulation or astrocytic signaling to plasticity and degeneration, would improve impact.

2, The section transitions are abrupt. The text oscillates between descriptive and historical narrative (e.g., “In 2005… In 2010…”) rather than thematic organization (e.g., “Mechanisms of tripartite synapse signaling,” “EV-mediated intercellular communication”).

3, The review lacks molecular depth in discussing the following aspects: the pathways linking astrocyte-secreted glypicans, neurocan, or TGF-β signaling to synapse formation; how EV cargo (miRNAs, tau, Arc) mechanistically alters neuronal connectivity or pathology and the intracellular mechanisms of EV biogenesis and release under physiological vs pathological conditions. The authors can include diagrams summarizing key mechanisms (e.g., EV-mediated synapse modulation) to enhance the discussion.

4, The discussion of AD and other neurodegenerative disorders is too general. While many references are cited, the manuscript does not critically evaluate how EVs or astrocytic factors causally contribute to pathology (e.g., tau propagation, amyloid clearance, synaptic pruning defects). Moreover, therapeutic discussions are cursory and lack mechanistic linkage.

5, Most conclusions reiterate established knowledge (“astrocytes regulate synapse formation,” “EVs mediate communication”). The paper would benefit from a forward-looking section proposing unresolved questions or new conceptual models, for instance, how EV-mediated signaling integrates with synaptic plasticity or liquid–liquid phase separation (LLPS) phenomena.

6, some specific comments. The concept of cleft expansion during stimulation in Figure 1 is interesting but speculative; the physiological evidence and implications for synaptic efficacy should be clarified. The table 1 repeats content from the main text without adding analytical value. It could instead summarize key astrocyte-derived factors, their receptors, and functional consequences. In Section 4 (Mechanisms to Diseases), this section merges multiple ideas (LLPS, exocytosis, neurodegeneration) with minimal mechanistic connection; these should be separated and supported by schematic models.  In Therapeutic Subsection, mentions of polyphenols and EV-based therapies would benefit from quantitative data (e.g., outcomes from animal or clinical studies).

Minor defects:

1, The English requires significant editing for conciseness and scientific precision.

2, The frequent repetition of phrases (“presented in this section…”, “the properties of these synapses…”) and inconsistent terminology (e.g., “thickness of clefts” vs “cleft enlargement”) make the manuscript verbose.

3, The figure captions are descriptive rather than explanatory and do not convey mechanistic insight.

Comments on the Quality of English Language

The English requires significant editing for conciseness and scientific precision.

Author Response

In the opinion of the third Reviewer, my contribution to the present review Is of considerable importance, however it lacks analytical depth, mechanistic integrations, and conceptual coherences. In my opinion this review, being member of a Special Issue, cannot cover all the properties and complexities that are considered at the moment. Rather, its evaluation is not superficial, it does not introduce all aspects of the literature. Upon its focused introduction, it does illustrate the main developments about synapses, appeared during the last few years.   
  1. The manuscript does not “reiterate prior findings”. It is mostly focused on recent developments coordinated in a general interpretation. The introduction of additional, very detailed aspects, i.e. the well known mechanistic details of EV properties that I have covered in my previous publications, are present now in the Reference list. The detailed presentation of all properties would require an unnecessary enlargement of our presentation.

  1.  My impression about the criticized issues is different, the changes (called oscillations, not repeated several times) have been used initially by introducing problems developed by recent studies.

  1. Several detailed aspects of the present state of synapses are missing in my present review, even if their several details result “strategical changes” of the present work. In fact, their presence in my review would need the development of multiple detailed presentations. Interesting, at least some of these suggestions could be considered for additional problems to be investigated in the next few years.

  1. From a different point of view, the suggested presentations, much more detailed than the one presented here, would require a new, mostly clinical review.

  1.  The suggestion of a benefit looking aspect, dealing with phenomena of synaptic plasticity or liquid–liquid phase separation, phenol-synaptic plasticity, or liquid–liquid phase separation (LLPS), is a process discovered a couple of years ago, that had been left out of my review. Based on the suggestion of the Reviewer, I have changed my decision by introducing  a paradigm of page 5 with 3 recent publications, the 43-45 references.
  2. I appreciated several specific comments received. Some have been considered in the revision of the text, the others for the developments of future work to be done.

Round 2

Reviewer 2 Report

Comments and Suggestions for Authors

From the letter of response I received, I couldn't quite discern the author's intent. Glancing over the main text, I thought they might have opted for minor revisions. However, since I originally only pointed out concerns as suggestions, I think it's fine if the author is satisfied with this revision.

Author Response

This Reviewer evaluated the revised version of the paper, the one corrected based on the critical revisions of Reviewer 2. The changes introduced following the criticisms of Reviewer 2 and the ensuing corrections introduced in the text were appropriate. On the other hand, some possible changes of re-revision could not be excluded, however they were only marginal, therefore unnecessary. Therefore, in the opinion on this Reviewer the present version of the paper should be approved and directly indicated for publication.

Reviewer 3 Report

Comments and Suggestions for Authors

The authors have extensively revised the article which render for further detailed review. I realize that the section 3: Astrocytes and other glial cells of glial cells need to add more discussion. The interactions between glial cells and synapse has not been fully covered. The recent Cell paper:  Benoit, L., et al. (2025). Astrocytes functionally integrate multiple synapses via specialized leaflet domains. Cell, 188, 6453–6472, should be cited and discussed for novel information. 

Round 3

Reviewer 3 Report

Comments and Suggestions for Authors

The author has not added new discussion for LLPS and microglial. The author only added some words related to LLPS and microglial in abstract section. No point-by-point response has been provided by the authors. The new Cell paper suggested by the reviewer has not been cited by the authors. So I can not suggest accepting of it.

Major revisions are still needed for this article.

Author Response

Upon his/her evaluation, the first revised version of the paper was found greatly improved, In particular by the introduction of two important changes concerning LLPS and microglia. These changes were really critical because they were focused on two aspects of synaptic properties that in the original version of the paper had been treated only marginally.

In the opinion of the Reviewer these limitations were not only present in the initial version of the paper. Rather, some other examples could be considered. In particular, the Reviewer emphasized a critical aspect of synapses, the general structure of astrocytes and their mechanisms of key importance for the pre- post-synaptic steps.

This situation has been greatly improved by the appearance in the Journal Cell by innovative results dependent on new techniques and interpretations by the paper of Benoit and many other scientists specialist of the field. Their collaboration has permitted me to change an important general strategy of my paper. In the initial version the data had been all reported in the Section 4, including Mechanisms, Diseases ad Therapies. In my re-revision I have divided the latter Section in two. The today’s Section 4 includes now the critical Mechanisms, in sequence the LLPS, the exocytoses of VSs and their regulation of pre- post-synapses by specialized leaflets domains. The results located or nerar to this Section have finally resulted in 18 more articles included in our References. The ensuing Section, now number 5, now includes Diseases and Therapies.

The changes of the paper and its presentations, including its details and other aspects, i.e. Table 1 and final Conclusions, have now increased their excellent level reached. I hope my re-revised version does not need any additional changes and therefore it will be considered positively for publication soon.    

Round 4

Reviewer 3 Report

Comments and Suggestions for Authors

This version has been significantly improved. No further concerns. I suggest accepting it

Author Response

-